# On Model Parallelization and Scheduling Strategies for Distributed Machine Learning

†Seunghak Lee, †Jin Kyu Kim, †Xun Zheng, §Qirong Ho, †Garth A. Gibson, †Eric P. Xing

| | |
|---|---|
| †School of Computer Science | §Institute for Infocomm Research |
| Carnegie Mellon University | A*STAR |
| Pittsburgh, PA 15213 | Singapore 138632 |
| seunghak@, jinkyuk@, xunzheng@, | hoqirong@gmail.com |
| garth@, epxing@cs.cmu.edu | |

## Abstract

Distributed machine learning has typically been approached from a *data parallel* perspective, where big data are partitioned to multiple workers and an algorithm is executed concurrently over different data subsets under various synchronization schemes to ensure speed-up and/or correctness. A sibling problem that has received relatively less attention is how to ensure efficient and correct *model parallel* execution of ML algorithms, where parameters of an ML program are partitioned to different workers and undergone concurrent iterative updates. We argue that model and data parallelisms impose rather different challenges for system design, algorithmic adjustment, and theoretical analysis. In this paper, we develop a system for model-parallelism, STRADS, that provides a programming abstraction for scheduling parameter updates by discovering and leveraging changing structural properties of ML programs. STRADS enables a flexible tradeoff between scheduling efficiency and fidelity to intrinsic dependencies within the models, and improves memory efficiency of distributed ML. We demonstrate the efficacy of model-parallel algorithms implemented on STRADS versus popular implementations for topic modeling, matrix factorization, and Lasso.

## 1 Introduction

Advancements in sensory technologies and digital storage media have led to a prevalence of "Big Data" collections that have inspired an avalanche of recent efforts on "scalable" machine learning (ML). In particular, numerous *data-parallel* solutions from both algorithmic [28, 10] and system [7, 25] angles have been proposed to speed up inference and learning on Big Data. The recently emerged *parameter server* architecture [15, 18] has started to pave ways for a unified programming interface for data parallel algorithms, based on various parallellization models such as stale synchronous parallelism (SSP) [15], eager SSP [5], and value-bound asynchronous parallelism [23], etc. However, in addition to Big Data, modern large-scale ML problems have started to encounter the so-called *Big Model* challenge [8, 1, 17], in which models with millions if not billions of parameters and/or variables (such as in deep networks [6] or large-scale topic models [20]) must be estimated from big (or even modestly-sized) datasets. Such Big Model problems seem to have received less systematic investigation. In this paper, we propose a *model-parallel* framework for such an investigation.

As is well known, a data-parallel algorithm parallelly computes a *partial* update of **all model parameters** (or latent model states in some cases) in each worker, based on only the subset of data on that worker and a local copy of the model parameters stored on that worker, and then aggregates these partial updates to obtain a global estimate of the model parameters [15]. In contrast, a model

parallel algorithm aims to parallelly update **a subset of parameters** on each worker — using either all data, or different subsets of the data [4] — in a way that preserves as much correctness as possible, by ensuring that the updates from each subset are highly compatible. Obviously, such a scheme directly alleviates memory bottlenecks caused by massive parameter sizes in big models; but even for small or mid-sized models, an effective model parallel scheme is still highly valuable because it can speed up an algorithm by updating multiple parameters concurrently, using multiple machines.

While data-parallel algorithms such as stochastic gradient descent [27] can be advantageous over their sequential counterparts — thanks to concurrent processing over data using various bounded-asynchronous schemes — they require every worker to have full access to all global parameters; furthermore they leverage on an assumption that different data subsets are *i.i.d.* given the shared global parameters. For a model-parallel program however, in which model parameters are distributed to different workers, one cannot blindly leverage such an *i.i.d.* assumption over arbitrary parameter subsets, because doing so will cause incorrect estimates due to incompatibility of sub-results from different workers (e.g., imagine trivially parallelizing a long, simplex-constrained vector across multiple workers — independent updates will break the simplex constraint). Therefore, existing data-parallel schemes and frameworks, that cannot support sophisticated constraint and/or consistency satisfiability mechanisms across workers, are not easily adapted to model-parallel programs. On the other hand, as explored in a number of recent works, explicit analysis of dependencies across model parameters, coupled with the design of suitable parallel schemes accordingly, opens up new opportunities for big models. For example, as shown in [4], model-parallel coordinate descent allows us to update multiple parameters in parallel, and our work in this paper further this approach by allowing some parameters to be prioritized over others. Furthermore, one can take advantage of model structures to avoid interference and loss of correctness during concurrent parameter updates (e.g., nearly independent parameters can be grouped to be updated in parallel [21]), and in this paper, we explore how to discover such structures in an efficient and scalable manner. To date, model-parallel algorithms are usually developed for a specific application such as matrix factorization [10] or Lasso [4] — thus, there is a need for developing programming abstractions and interfaces that can tackle the common challenges of Big Model problems, while also exposing new opportunities such as parameter prioritization to speed up convergence without compromising inference correctness.

Effectively and conveniently programming a model-parallel algorithm stands as another challenge, as it requires mastery of detailed communication management in a cluster. Existing distributed frameworks such as MapReduce [7], Spark [25], and GraphLab [19] have shown that a variety of ML applications can be supported by a single, common programming interface (e.g. Map/Reduce or Gather/Apply/Scatter). Crucially, these frameworks allow the user to specify a coarse order to parameter updates, but automatically decide on the precise execution order — for example, MapReduce and Spark allow users to specify that parallel jobs should be executed in some topological order; e.g. mappers are guaranteed to be followed by reducers, but the system will execute the mappers in an arbitrary parallel or sequential order that it deems suitable. Similarly, GraphLab chooses the next node to be updated based on its "chromatic engine" and the user's choice of graph consistency model, but the user only has loose control over the update order (through the input graph structure). While this coarse-grained, fully-automatic scheduling is certainly convenient, it does not offer the fine-grained control needed to avoid parallelization of parameters with subtle interdependencies that might not be present in the superficial problem or graph structure (which can then lead to algorithm divergence, as in Lasso [4]). Moreover, most of these frameworks do not allow users to easily prioritize parameters based on new criteria, for more rapid convergence (though we note that GraphLab allows node prioritization through a priority queue). It is true that data-parallel algorithms can be implemented efficiently on these frameworks, and in principle, one can also implement model-parallel algorithms on top of them. Nevertheless, we argue that without fine-grained control over parameter updates, we would miss many new opportunities for accelerating ML algorithm convergence.

To address these challenges, we develop STRADS (STRucture-Aware Dynamic Scheduler), a system that performs automatic scheduling and parameter prioritization for dynamic Big Model parallelism, and is designed to enable investigation of new ML-system opportunities for efficient management of memory and accelerated convergence of ML algorithms, while making a best-effort to preserve existing convergence guarantees for model-parallel algorithms (e.g. convergence of Lasso under parallel coordinate descent). STRADS provides a simple abstraction for users to program ML algorithms, consisting of three "conceptual" actions: **schedule**, **push** and **pull**. **Schedule** specifies the next subset of model parameters to be updated in parallel, **push** specifies how individual workers

compute partial results on those parameters, and **pull** specifies how those partial results are aggregated to perform the full parameter update. A high-level view of STRADS is illustrated in Figure 1. We stress that these actions only specify the abstraction for managed model-parallel ML programs; *they do not dictate the underlying implementation*. A key-value store allows STRADS to handle a large number of parameters in distributed fashion, accessible from all master and worker machines.

As a showcase for STRADS, we implement and provide **schedule/push/pull** pseudocode for three popular ML applications: topic modeling (LDA), matrix factorization (MF), and Lasso. It is our hope that: (1) the STRADS interface enables Big Model problems to be solved in distributed fashion with modest programming effort, and (2) the STRADS mechanism accelerates the convergence Big ML algorithms through good scheduling (particularly through used-defined scheduling

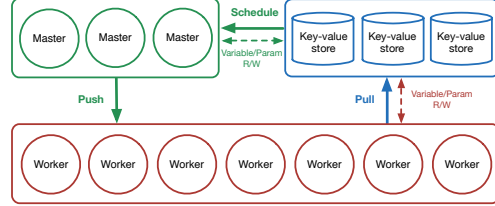

Figure 1: High-level architecture of our STRADS system interface for dynamic model parallelism.

criteria). In our experiments, we present some evidence of STRADS's success: topic modeling with 3.9M docs, 10K topics, and 21.8M vocabulary (200B parameters), MF with rank-2K on a 480K-by-10K matrix (1B parameters), and Lasso with 100M features (100M parameters).

## 2  Scheduling for Big Model Parallelism with STRADS

"Model parallelism" refers to parallelization of an ML algorithm over the space of shared model parameters, rather than the space of (usually i.i.d.) data samples. At a high level, model parameters are the changing intermediate quantities that an ML algorithm iteratively updates, until convergence is reached. A key advantage of the model-parallel approach is that it explicitly partitions the model parameters into subsets, allowing ML problems with massive model spaces to be tackled on machines with limited memory (see supplement for details of STRADS memory usage).

To enable users to systematically and programmatically exploit model parallelism, STRADS defines a programming interface, where the user writes three functions for a ML problem: **schedule**, **push** and **pull** (Figures 1, 2). STRADS repeatedly schedules

```
// Generic STRADS application
```

```
schedule() {
  // Select U params x[j] to be sent
  // to the workers for updating
  ...
  return (x[j_1], ..., x[j_U])
}
```

```
push(worker = p, pars = (x[j_1],...,x[j_U])) {
  // Compute partial update z for U params x[j]
  // at worker p
  ...
  return z
}
```

```
pull(workers = [p], pars = (x[j_1],...,x[j_U]),
     updates = [z]) {
  // Use partial updates z from workers p to
  // update U params x[j]. sync() is automatic.
  ...
}
```

Figure 2: **STRADS interface:** Basic functional signatures of **schedule**, **push**, **pull**, using pseudocode.

and executes these functions in that order, thus creating an iterative model-parallel algorithm. Below, we describe the three functions.

**Schedule:**  This function selects $U$ model parameters to be dispatched for updates (Figure 1). Within the **schedule** function, the programmer may access all data $D$ and all model parameters $x$, in order to decide which $U$ parameters to dispatch. A simple **schedule** is to select model parameters according to a fixed sequence, or drawn uniformly at random. As we shall later see, **schedule** also allows model parameters to be selected in a way that: (1) focuses on the fastest-converging parameters, while avoiding already-converged parameters; (2) avoids parallel dispatch of parameters with inter-dependencies, which can lead to divergence or parallelization errors.

**Push & Pull:**  These functions describe the flow of model parameters $x$ from the scheduler to the workers performing the update equations, as in Fig 1. **Push** dispatches a set of parameters $\{x_{j_1}, \ldots, x_{j_U}\}$ to each worker $p$, which then computes a partial update $z$ for $\{x_{j_1}, \ldots, x_{j_U}\}$ (or a subset of it). When writing **push**, the user can take advantage of data partitioning: e.g., when only a fraction $\frac{1}{P}$ of the data samples are stored at each worker, the $p$-th worker should compute partial results $z_j^p = \sum_{D_i} f_{x_j}(D_i)$ by iterating over its $\frac{1}{P}$ data points $D_i$. **Pull** is used to collect the partial results $\{z_j^p\}$ from all workers, and commit them to the parameters $\{x_{j_1}, \ldots, x_{j_U}\}$. Our STRADS LDA, MF, and Lasso applications partition the data samples uniformly over machines.

# 3 Leveraging Model-Parallelism in ML Applications through STRADS

In this section, we explore how users can apply model-parallelism to their ML applications, using STRADS. As case studies, we design and experiment on 3 ML applications — LDA, MF, and Lasso — in order to show that model-parallelism in STRADS can be simple to implement, yet also powerful enough to expose new and interesting opportunities for speeding up distributed ML.

## 3.1 Latent Dirichlet Allocation (LDA)

We introduce STRADS programming through topic modeling via LDA [3]. Big LDA models provide a strong use case for model-parallelism: when thousands of topics and millions of words are used, the LDA model contains billions of global parameters, and data-parallel implementations face the challenge of providing access to all these parameters; in contrast, model-parallellism explicitly divides up the parameters, so that workers only need to access a fraction of parameters at a given time.

Formally, LDA takes a corpus of $N$ documents as input — represented as word "tokens" $w_{ij} \in \boldsymbol{W}$, where $i$ is the document index and $j$ is the word position index — and outputs $K$ topics as well as $N$ $K$-dimensional topic vectors (soft assignments of topics to each document). LDA is commonly reformulated as a "collapsed" model [14], in which some of the

```
// STRADS LDA

schedule() {
  dispatch = []  // Empty list
  for a=1..U     // Rotation scheduling
    idx = ((a+C-1) mod U) + 1
    dispatch.append( V[q_idx] )
  return dispatch
}

push(worker = p, pars = [V_a, ..., V_U]) {
  t = []                   // Empty list
  for (i,j) in W[q_p]  // Fast Gibbs sampling
    if w[i,j] in V_p
      t.append( (i,j,f_1(i,j,D,B)) )
  return t
}

pull(workers = [p], pars = [V_a, ..., V_U],
      updates = [t]) {
  for all (i,j)     // Update sufficient stats
    (D,B) = f_2([t])
}
```

Figure 3: **STRADS LDA pseudocode.** Definitions for $f_1, f_2, q_p$ are in the text. C is a global model parameter.

latent variables are integrated out for faster inference. Inference is performed using Gibbs sampling, where each word-topic indicator (denoted $z_{ij} \in \boldsymbol{Z}$) is sampled in turn according to its distribution conditioned on all other parameters. To perform this computation without having to iterate over all $\boldsymbol{W}, \boldsymbol{Z}$, sufficient statistics are kept in the form of a "doc-topic" table $\boldsymbol{D}$, and a "word-topic" table $\boldsymbol{B}$. A full description of the LDA model is in the supplement.

**STRADS implementation:** In order to perform model-parallelism, we first identify the model parameters, and create a **schedule** strategy over them. In LDA, the assignments $z_{ij}$ are the model parameters, while $\boldsymbol{D}, \boldsymbol{B}$ are summary statistics over $z_{ij}$ that are used to speed up the sampler. Our **schedule** strategy equally divides the $V$ words into $U$ subsets $V_1, \ldots, V_U$ (where $U$ is the number of workers). Each worker will only sample words from one subset $V_a$ at a time (via **push**), and update the sufficient statistics $\boldsymbol{D}, \boldsymbol{W}$ via **pull**. Subsequent invocations of **schedule** will "rotate" subsets amongst workers, so that every worker touches all $U$ subsets every $U$ invocations. For data partitioning, we divide the document tokens $w_{ij} \in \boldsymbol{W}$ evenly across workers, and denote worker $p$'s set of tokens by $\boldsymbol{W}_{q_p}$, where $q_p$ is the index set for the

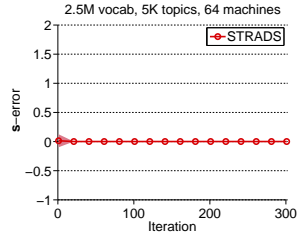

Figure 4: **STRADS LDA:** Parallelization error $\Delta_t$ at each iteration, on the Wikipedia unigram dataset with $K = 5000$ and 64 machines.

$p$-th worker. Further details and analysis of the pseudocode, particularly how **push**-**pull** constitutes a model-parallel execution of LDA, are in the supplement.

**Model parallelism results in low error:** Parallel Gibbs sampling is not generally guaranteed to converge [12], unless the parameters being sampled for concurrent updates are conditionally independent of each other. STRADS model-parallel LDA assigns workers to *disjoint* words $V$ and documents $w_{ij}$; thus, each worker's parameters $z_{ij}$ are almost conditionally independent of other workers, resulting in very low sampling error [1]. As evidence, we define an error score $\Delta_t$ that measures the divergence between the true word-topic distribution/table $\boldsymbol{B}$, versus the local copy seen at each worker (a full mathematical explanation is in the supplement). $\Delta_t$ ranges from $[0, 2]$ (where 0 means no error). Figure 4 plots $\Delta_t$ for the "Wikipedia unigram" dataset (see §5 for

experimental details) with $K = 5000$ topics and 64 machines (128 processor cores total). $\Delta_t$ is $\leq 0.002$ throughout, confirming that STRADS LDA exhibits very small parallelization error.

## 3.2 Matrix Factorization (MF)

We now consider matrix factorization (collaborative filtering), which can be used to predict users' unknown preferences, given their known preferences and the preferences of others. Formally, MF takes an incomplete matrix $\mathbf{A} \in \mathbb{R}^{N \times M}$ as input, where $N$ is the number of users, and $M$ is the number of items. The idea is to discover rank-$K$ matrices $\mathbf{W} \in \mathbb{R}^{N \times K}$ and $\mathbf{H} \in \mathbb{R}^{K \times M}$ such that $\mathbf{WH} \approx \mathbf{A}$. Thus, the product $\mathbf{WH}$ can be used to predict the missing entries (user preferences). Let $\Omega$ be the set of indices of observed entries in $\mathbf{A}$, let $\Omega^i$ be the set of observed column indices in the $i$-th row of $\mathbf{A}$, and let $\Omega_j$ be the set of observed row indices in the $j$-th column of $\mathbf{A}$. Then, the MF task is defined by an optimization problem: $\min_{\mathbf{W},\mathbf{H}} \sum_{(i,j) \in \Omega} (a_j^i - \mathbf{w}^i \mathbf{h}_j)^2 + \lambda(\|\mathbf{W}\|_F^2 + \|\mathbf{H}\|_F^2)$. We solve this objective using a parallel coordinate descent algorithm [24].

**STRADS implementation:** Our MF **schedule** strategy is to partition the rows of $\mathbf{A}$ into $U$ disjoint index sets $q_p$, and the columns of $\mathbf{A}$ into $U$ disjoint index sets $r_p$. We then dispatch the model parameters $\mathbf{W}, \mathbf{H}$ in a round-robin

```
// STRADS Matrix Factorization

schedule() {
  // Round-robin scheduling
  if counter <= U      // Do W
    return W[q_counter]
  else                 // Do H
    return H[r_(counter-U)]
}
```

```
push(worker = p, pars = X[s]) {
  z = []              // Empty list
  if counter <= U   // X is from W
    for row in s, k=1..K
      z.append( (f_1(row,k,p),f_2(row,k,p)) )
  else                // X is from H
    for col in s, k=1..K
      z.append( (g_1(k,col,p),g_2(k,col,p)) )
  return z
}
```

```
pull(workers=[p], pars=X[s], updates=[z]) {
  if counter <= U    // X is from W
    for row in s, k=1..K
      W[row,k] = f_3(row,k,[z])
  else                // X is from H
    for col in s, k=1..K
      H[k,col] = g_3(k,col,[z])
  counter = (counter mod 2*U) + 1
}
```

Figure 5: **STRADS MF pseudocode.** Definitions for $f_1, g_1, \ldots$ and $q_p, r_p$ are in the text. `counter` is a global model variable.

fashion. To update the rows of $\mathbf{W}$, each worker $p$ uses **push** to compute partial summations on its assigned columns $r_p$ of $\mathbf{A}$ and $\mathbf{H}$; the columns of $\mathbf{H}$ are updated similarly with rows $q_p$ of $\mathbf{A}$ and $\mathbf{W}$. Finally, **pull** aggregates the partial summations, and then update the entries in $\mathbf{W}$ and $\mathbf{H}$. In Figure 5, we show the STRADS MF pseudocode, and further details are in the supplement.

## 3.3 Lasso

STRADS not only supports simple static **schedules**, but also dynamic, adaptive strategies that take the model state into consideration. Specifically, STRADS Lasso implementation schedules parameter updates by (1) prioritizing coefficients that contribute the most to algorithm convergence, and (2) avoiding the simultaneous update of coefficients whose dimensions are highly inter-dependent. These properties complement each other in an algorithmically efficient way, as we shall see.

Formally, Lasso can be defined by an optimization problem: $\min_{\boldsymbol{\beta}} \frac{1}{2} \|\mathbf{y} - \mathbf{X}\boldsymbol{\beta}\|_2^2 + \lambda \sum_j |\beta_j|$, where $\lambda$ is a regularization parameter that determines the sparsity of $\boldsymbol{\beta}$. We solve Lasso using coordinate descent (CD) update rule [9]: $\beta_j^{(t)} \leftarrow S(\mathbf{x}_j^T \mathbf{y} - \sum_{j \neq k} \mathbf{x}_j^T \mathbf{x}_k \beta_k^{(t-1)}, \lambda)$, where $S(g, \lambda) := \text{sign}(\beta) (|g| - \lambda)_+$.

**STRADS implementation:** Lasso **schedule** dynamically selects parameters to be updated with the following prioritization scheme: rapidly changing parameters are more frequently updated than others. First, we define a probability distribution $\mathbf{c} = [c_1, \ldots, c_J]$ over $\boldsymbol{\beta}$; the purpose of $\mathbf{c}$ is to prioritize $\beta_j$'s during **schedule**, and thus speed up convergence. In particular, we observe that choosing $\beta_j$ with probability $c_j = f_1(j) :\propto \left(\delta\beta_j^{(t-1)}\right)^2 + \eta$ substantially speeds up the Lasso convergence rate, where $\eta$ is a small positive constant, and $\delta\beta_j^{(t-1)} = \beta_j^{(t-2)} - \beta_j^{(t-1)}$.

To prevent non-convergence due to dimension inter-dependencies [4], we only **schedule** $\beta_j$ and $\beta_k$ for concurrent updates if $\mathbf{x}_j^T \mathbf{x}_k \approx 0$. This is performed as follows: first, select $L'(> L)$ indices of coefficients from the probability distribution $\mathbf{c}$ to form a set $\mathcal{C}$ ($|\mathcal{C}| = L'$). Next, choose a subset $\mathcal{B} \subset \mathcal{C}$ of size $L$ such that $\mathbf{x}_j^T \mathbf{x}_k < \rho$ for all $j, k \in \mathcal{B}$, where $\rho \in (0, 1)$; we represent this selection procedure by the function $f_2(\mathcal{C})$. Note that this procedure is inexpensive: by selecting $L'$ candidate

$\beta_j$'s first, only $L'^2$ dependencies need to be checked, as opposed to $J^2$, where $J$ is the total number of features. Here $L'$ and $\rho$ are user-defined parameters.

We execute **push** and **pull** to update the coefficients indexed by $\mathcal{B}$ using $U$ workers in parallel. The rows of the data matrix $\mathbf{X}$ are partitioned into $U$ submatrices, and the $p$-th worker stores the submatrix $\mathbf{X}_{q_p} \in \mathbb{R}^{|q_p| \times J}$; with $\mathbf{X}$ partitioned in this manner, we need to modify the CD update rule accordingly. Using $U$ workers, **push** computes $U$ partial summations for each selected $\beta_j, j \in \mathcal{B}$, denoted by $\{z_{j,1}^{(t)}, \ldots, z_{j,U}^{(t)}\}$, where $z_{j,p}$ represents the partial summation for $\beta_j$ in the $p$-th worker at the $t$-th iteration: $z_{j,p}^{(t)} \leftarrow f_3(p, j) := \sum_{i \in q_p} \left\{ (\mathbf{x}_j^i)^T \mathbf{y} - \sum_{j \neq k} (\mathbf{x}_j^i)^T (\mathbf{x}_k^i) \beta_k^{(t-1)} \right\}$. After all **pushes** have been completed, **pull** updates $\beta_j$ via $\beta_j^{(t)} = f_4(j, [z_{j,p}^{(t)}]) := S(\sum_{p=1}^{U} z_{j,p}^{(t)}, \lambda)$.

**Analysis of STRADS Lasso scheduling** We wish to highlight several notable aspects of the STRADS Lasso **schedule** mentioned above. In brief, the sampling distribution $f_1(j)$ and the model dependency control scheme with threshold $\rho$ allow STRADS to speed up the convergence rate of Lasso. To analyze this claim, let us rewrite the Lasso problem by duplicating original features with opposite sign: $F(\boldsymbol{\beta}) := \min_{\boldsymbol{\beta}} \frac{1}{2} \|\mathbf{y} - \mathbf{X}\boldsymbol{\beta}\|_2^2 + \lambda \sum_{j=1}^{2J} \beta_j$. Here, with an abuse of notation, $\mathbf{X}$ contains $2J$ features and $\beta_j \geq 0$, for all $j = 1, \ldots, 2J$. Then, we have the following analysis of our scheduling scheme.

```
// STRADS Lasso

schedule() {
  // Priority-based scheduling
  for all j      // Get new priorities
    c_j = f_1(j)
  for a=1..L'    // Prioritize betas
    random draw s_a using [c_1, ..., c_J]
  // Get 'safe' betas
  (j_1, ..., j_L) = f_2(s_1, ..., s_L')
  return (b[j_1], ..., b[j_L])
}
```

```
push(worker = p, pars = (b[j_1],...,b[j_L])) {
  z = []          // Empty list
  for a=1..L     // Compute partial sums
    z.append( f_3(p,j_a) )
  return z
}
```

```
pull(workers = [p], pars = (b[j_1],...,b[j_L]),
     updates = [z]) {
  for a=1..L          // Aggregate partial sums
    b[j_a] = f_4(j_a,[z])
}
```

Figure 6: **STRADS Lasso pseudocode.** Definitions for $f_1, f_2, \ldots$ are given in the text.

**Proposition 1** *Suppose $\mathcal{B}$ is the set of indices of coefficients updated in parallel at the $t$-th iteration, and $\rho$ is sufficiently small constant such that $\rho \delta \beta_j^{(t)} \delta \beta_k^{(t)} \approx 0$, for all $j \neq k \in \mathcal{B}$. Then, the sampling distribution $p(j) \propto \left(\delta \beta_j^{(t)}\right)^2$ approximately maximizes a lower bound on $E_{\mathcal{B}} \left[ F(\boldsymbol{\beta}^{(t)}) - F(\boldsymbol{\beta}^{(t)} + \Delta \boldsymbol{\beta}^{(t)}) \right]$.*

Proposition 1 (see supplement for proof) shows that our scheduling attempts to speed up the convergence of Lasso by decreasing the objective as much as possible at every iteration. However, in practice, we approximate $p(j) \propto \left(\delta \beta_j^{(t)}\right)^2$ with $f_1(j) \propto \delta \left(\beta_j^{(t-1)}\right)^2 + \eta$ because $\delta \beta_j^{(t)}$ is unavailable at the $t$-th iteration before computing $\beta_j^{(t)}$; we add $\eta$ to give all $\beta_j$'s non-zero probability of being updated to account for the approximation.

# 4   STRADS System Architecture and Implementation

Our STRADS system implementation uses multiple master/scheduler machines, multiple worker machines, and a single "master" coordinator[2] machine that directs the activities of the schedulers and workers The basic unit of STRADS execution is a "round", which consists of **schedule-push-pull** in that order. In more detail (Figure 1), (1) the masters execute **schedule** to pick $U$ sets of model parameters $x$ that can be safely updated in parallel (if the masters need to read parameters, they get them from the key-value stores); (2) jobs for **push**, which update the $U$ sets of parameters, are dispatched via the coordinator to the workers (again, workers read parameters from the key-value stores), which then execute **push** to compute partial updates $z$ for each parameter; (3) the key-value stores execute **pull** to aggregate the partial updates $z$, and keep newly updated parameters.

To efficiently use multiple cores/machines in the scheduler pool, STRADS uses pipelined **schedule** computations, i.e., masters compute **schedule** and queue jobs in advance for future rounds. In other

words, parameters to be updated are determined by the masters without waiting for workers' parameter updates; the jobs for parameter updates are dispatched to workers in turn by the coordinator. By pipelining **schedule**, the master machines do not become a bottleneck even with a large number of workers. Specifically, the pipelined strategy does not occur any parallelization errors if parameters $x$ for **push** can be ordered in a manner that does not depend on their actual values (e.g. MF and LDA applications). For programs whose **schedule** outcome depends on the current values of $x$ (e.g. Lasso), the strategy is equivalent to executing **schedule** based on *stale* values of $x$, similar to how parameter servers allow computations to be executed on stale model parameters [15, 1]. In Lasso experiments in §5, such **schedule** strategy with stale values greatly improved its convergence rate.

STRADS does not have to perform **push-pull** communication between the masters and the workers (which would bottleneck the masters). Instead, the model parameters $x$ can be globally accessible through a distributed, partitioned key-value store (represented by standard arrays in our pseudocode). A variety of key-value store synchronization schemes exist, such as Bulk Synchronous Parallel (BSP), Stale Synchronous Parallel (SSP) [15], and Asynchronous Parallel (AP). In this paper, we use BSP synchronization; we leave the use of alternative schemes like SSP or AP as future work. We implemented STRADS using C++ and the Boost libraries, and OpenMPI 1.4.5 was used for asynchronous communication between the master schedulers, workers, and key-value stores.

## 5   Experiments

We now demonstrate that our STRADS implementations of LDA, MF and Lasso can (1) reach larger model sizes than other baselines; (2) converge at least as fast, if not faster, than other baselines; (3) with additional machines, STRADS uses less memory per machine (efficient partitioning). For baselines, we used (a) a STRADS implementation of distributed Lasso with only a naive round-robin scheduler (Lasso-RR), (b) GraphLab's Alternating Least Squares (ALS) implementation of MF [19], (c) YahooLDA for topic modeling [1]. Note that Lasso-RR imitates the random scheduling scheme proposed by Shotgun algorithm on STRADS. We chose GraphLab and YahooLDA, as they are popular choices for distributed MF and LDA.

We conducted experiments on two clusters [11] (with 2-core and 16-core machines respectively), to show the effectiveness of STRADS model-parallelism across different hardware. We used the 2-core cluster for LDA, and the 16-core cluster for Lasso and MF. The 2-core cluster contains 128 machines, each with two 2.6GHz AMD cores and 8GB RAM, and connected via a 1Gbps network interface. The 16-core cluster contains 9 machines, each with 16 2.1GHz AMD cores and 64GB RAM, and connected via a 40Gbps network interface. Both clusters exhibit a 4GB memory-to-CPU ratio, a setting commonly observed in the machine learning literature [22, 13], which closely matches the more cost-effective instances on Amazon EC2. All our experiments use a fixed data size, and we vary the number of machines and/or the model size (unless otherwise stated); furthermore, for Lasso, we set $\lambda = 0.001$, and for MF, we set $\lambda = 0.05$.

### 5.1   Datasets

**Latent Dirichlet Allocation**   We used 3.9M English Wikipedia abstracts, and conducted experiments using both unigram (1-word) tokens ($V = 2.5$M unique unigrams, 179M tokens) and bigram (2-word) tokens [16] ($V = 21.8$M unique bigrams, 79M tokens). We note that our bigram vocabulary (21.8M) is an order of magnitude larger than recently published results [1], demonstrating that STRADS scales to very large models. We set the number of topics to $K = 5000$ and $10000$ (also larger than recent literature [1]), which yields extremely large word-topic tables: 25B elements (unigram) and 218B elements (bigram).

**Matrix Factorization**   We used the Nexflix dataset [2] for our MF experiments: 100M anonymized ratings from 480,189 users on 17,770 movies. We varied the rank of $\mathbf{W}, \mathbf{H}$ from $K = 20$ to $2000$, which exceeds the upper limit of previous MF papers [26, 10, 24].

**Lasso**   We used synthetic data with 50K samples and $J = 10$M to 100M features, where every feature $\mathbf{x}_j$ has only 25 non-zero samples. To simulate correlations between adjacent features (which exist in real-world data sets), we first generate $\mathbf{x}_1 \sim Unif(0, 1)$. Then, with 0.9 probability, we make $\mathbf{x}_j \sim Unif(0, 1)$, and with 0.1 probability, $\mathbf{x}_j \sim 0.9\mathbf{x}_{j-1} + 0.1 Unif(0, 1)$ for $j = 2, \ldots, J$.

### 5.2   Speed and Model Sizes

Figure 7 shows the time taken by each algorithm to reach a fixed objective value (over a range of model sizes), as well as the largest model size that each baseline was capable of running. For LDA and MF, STRADS handles much larger model sizes than either YahooLDA (could handle 5K topics

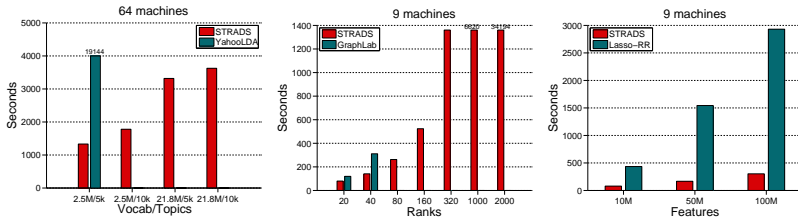

Figure 7: Convergence time versus model size for STRADS and baselines for (left) LDA, (center) MF, and (right) Lasso. We omit the bars if a method did not reach 98% of STRADS's convergence point (YahooLDA and GraphLab-MF failed at 2.5M-Vocab/10K-topics and rank $K \geq 80$, respectively). STRADS not only reaches larger model sizes than YahooLDA, GraphLab, and Lasso-RR, but also converges significantly faster.

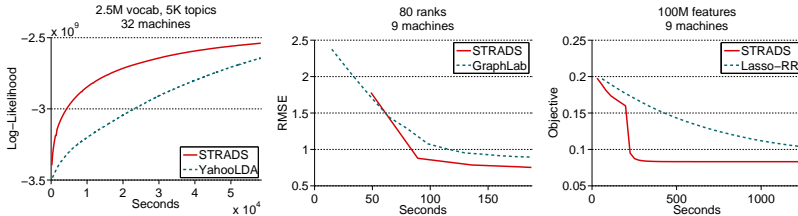

Figure 8: Convergence trajectories of different methods for (left) LDA, (center) MF, and (right) Lasso.

on the unigram dataset) or GraphLab (could handle rank $< 80$), while converging more quickly; we attribute STRADS's faster convergence to lower parallelization error (LDA only) and reduced synchronization requirements through careful model partitioning (LDA, MF). We observed that each YahooLDA worker stores a portion of the word-topic table — specifically, those elements referenced by the words in the worker's data partition. Because our experiments feature very large vocabulary sizes, even a small fraction of the word-topic table can still be too large for a single machine's memory, which caused YahooLDA to fail on the larger experiments. For Lasso, STRADS converges more quickly than Lasso-RR because of our dynamic **schedule** strategy, which is graphically captured in the convergence trajectory seen in Figure 8 — observe that STRADS's dynamic **schedule** causes the Lasso objective to plunge quickly to the optimum at around 250 seconds. We also see that STRADS LDA and MF achieved better objective values than the other baselines, confirming that STRADS model-parallelism is fast without compromising convergence quality.

## 5.3 Scalability

In Figure 9, we show the convergence trajectories and time-to-convergence for STRADS LDA using different numbers of machines at a fixed model size (unigram with 2.5M vocab and 5K topics). The plots confirm that STRADS LDA exhibits faster convergence with more machines, and that the time to convergence almost halves with every doubling of machines (near-linear scaling).

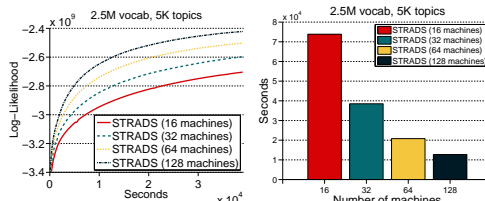

Figure 9: STRADS LDA scalability with increasing machines using a fixed model size. (Left) Convergence trajectories; (Right) Time taken to reach a log-likelihood of $-2.6 \times 10^9$.

## 6 Conclusions

In this paper, we presented a programmable framework for dynamic Big Model-parallelism that provides the following benefits: (1) scalability and efficient memory utilization, allowing larger models to be run with additional machines; (2) the ability to invoke dynamic **schedules** that reduce model parameter dependencies across workers, leading to lower parallelization error and thus faster, correct convergence. An important direction for future research would be to reduce the communication costs of using STRADS. We also want to explore the use of STRADS for other popular ML applications, such as support vector machines and logistic regression.

### Acknowledgments

This work was done under support from NSF IIS1447676, CNS-1042543 (PRObE [11]), DARPA FA87501220324, and support from Intel via the Intel Science and Technology Center for Cloud Computing (ISTC-CC).

## Footnotes

[1]This sampling error arises because workers see different versions $\boldsymbol{B}$ — which is an unavoidable when parallelizing LDA inference, because the Gibbs sampler is inherently sequential.

[2] The coordinator sends jobs from the masters and the workers, which does not bottleneck at the 10- to 100-machine scale explored in this paper. Distributing the coordinator is left for future work.

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
