[Supplementary Material]

# Supplement: On Model Parallelization and Scheduling Strategies for Distributed Machine Learning

†**Seunghak Lee**, †**Jin Kyu Kim**, †**Xun Zheng**, §**Qirong Ho**, †**Garth A. Gibson**, †**Eric P. Xing**

†School of Computer Science
Carnegie Mellon University
Pittsburgh, PA 15213
seunghak@, jinkyuk@, xunzheng@,
garth@, epxing@cs.cmu.edu

§Institute for Infocomm Research
A*STAR
Singapore 138632
hoqirong@gmail.com

## 1   Full STRADS Program for Latent Dirichlet Allocation (LDA)

Big topic models (using LDA [2]) provide a strong use case for model-parallelism: when thousands of topics and millions of words are used, the LDA model contains billions of global parameters, and data-parallel implementations face the difficult challenge of providing access to all these parameters; in contrast, model-parallellism explicitly divides up the parameters, so that workers only need to access a fraction at a given time.

Formally, LDA takes a corpus of $N$ documents as input, and outputs $K$ topics (each topic is just a categorical distribution over all $V$ unique words in the corpus) as well as $N$ $K$-dimensional topic vectors (soft assignments of topics to documents). The LDA model is

$$\mathrm{P}(\boldsymbol{W} \mid \boldsymbol{Z}, \boldsymbol{\theta}, \boldsymbol{\beta}) = \prod_{i=1}^{N} \prod_{j=1}^{M_i} \mathrm{P}(w_{ij} \mid z_{ij}, \boldsymbol{\beta}) \mathrm{P}(z_{ij} \mid \boldsymbol{\theta}_i),$$

where (1) $w_{ij}$ is the $j$-th token (word position) in the $i$-th document, (2) $M_i$ is the number of tokens in document $i$, (3) $z_{ij}$ is the topic assignment for $w_{ij}$, (4) $\boldsymbol{\theta}_i$ is the topic vector for document $i$, and (5) $\boldsymbol{\beta}$ is a matrix representing the $K$ $V$-dimensional topics. LDA is commonly reformulated as a "collapsed" model [6] in which $\boldsymbol{\theta}, \boldsymbol{\beta}$ are integrated out for faster inference. Inference is performed using Gibbs sampling, where each $z_{ij}$ is sampled in turn according to its distribution conditioned on all other parameters, $\mathrm{P}(z_{ij} \mid \boldsymbol{W}, \boldsymbol{Z}_{-ij})$. To perform this computation without having to iterate over all $\boldsymbol{W}, \boldsymbol{Z}$, sufficient statistics are kept in the form of a "doc-topic" table $\boldsymbol{D}$ (analogous to $\boldsymbol{\theta}$), and a "word-topic" table $\boldsymbol{B}$ (analogous to $\boldsymbol{\beta}$). More precisely, $D_{ik}$ counts the number of assignments $z_{ij} = k$ in doc $i$, while $B_{vk}$ counts the number of tokens $w_{ij} = v$ such that $z_{ij} = k$.

**STRADS implementation:**   In order to perform model-parallelism, we first identify the model parameters, and create a **schedule** strategy over them. In LDA, the assignments $z_{ij}$ are the model parameters, while $\boldsymbol{D}, \boldsymbol{B}$ are summary statistics over the $z_{ij}$ that are used to speed up the sampler. Our **schedule** strategy equally divides the $V$ words into $U$ subsets $V_1, \dots, V_U$ (where $U$ is the number of workers). Each worker will only process words from one subset $V_a$ at a time. Subsequent invocations of **schedule** will "rotate" subsets amongst workers, so that every worker touches all $U$ subsets every $U$ invocations. For data partitioning, we divide the document tokens $\boldsymbol{W}$ evenly across workers, and denote worker $p$'s set of tokens by $\boldsymbol{W}_{q_p}$.

During **push**, suppose that worker $p$ is assigned to subset $V_a$ by **schedule**. This worker will only Gibbs sample the topic assignments $z_{ij}$ such that (1) $(i, j) \in \boldsymbol{W}_{q_p}$ and (2) $w_{ij} \in V_a$. In other words, $w_{ij}$ must be assigned to worker $p$, and must also be a word in $V_a$. The latter condition is the source of model-parallelism: observe how the assignments $z_{ij}$ are chosen for sampling based on word divisions $V_a$. Note that all $z_{ij}$ will be sampled exactly once after $U$ invocations of **schedule**. We use the fast Gibbs sampler from [12] to **push** update $z_{ij} \leftarrow f_1(i, j, \boldsymbol{D}, \boldsymbol{B})$, where $f_1()$ represents

```
// STRADS LDA

schedule() {
  dispatch = []   // Empty list
  for a=1..U       // Rotation scheduling
    idx = ((a+C-1) mod U) + 1
    dispatch.append( V[q_idx] )
  return dispatch
}

push(worker = p, pars = [V_a, ..., V_U]) {
  t = []            // Empty list
  for (i,j) in W[q_p]  // Fast Gibbs sampling
    if w[i,j] in V_p
      t.append( (i,j,f_1(i,j,D,B)) )
  return t
}

pull(workers = [p], pars = [V_a, ..., V_U], updates = [t]) {
  for all (i,j)     // Update sufficient stats
    (D,B) = f_2([t])
}
```

Figure 1: **STRADS LDA pseudocode.** Definitions for $f_1, f_2, q_p$ are in the text. C is a global model parameter.

the fast Gibbs sampler equation. The **pull** step simply updates the sufficient statistics $\boldsymbol{D}, \boldsymbol{B}$ using the new $z_{ij}$, and we represent this procedure as a function $(\boldsymbol{D}, \boldsymbol{B}) \leftarrow f_2([z_{ij}])$. Figure 1 provides pseudocode for STRADS LDA.

**Model parallelism results in low error:** Parallel Gibbs sampling is not generally guaranteed to converge [5], unless the parameters being parallel-sampled are conditionally independent of each other. Because STRADS LDA assigns workers to disjoint words $V$ and documents $w_{ij}$, each worker's parameters $z_{ij}$ are (almost) conditionally independent of other workers, except for a single shared dependency: the column sums of $\boldsymbol{B}$ (denoted by $\boldsymbol{s}$, and stored as an extra row appended to $\boldsymbol{B}$), which are required for correct normalization of the Gibbs sampler conditional distributions in $f_1()$. The column sums $\boldsymbol{s}$ are **synced** at the end of every **pull**, but will go out-of-sync during worker **pushes**. To understand how error in $\boldsymbol{s}$ affects sampler convergence, consider the Gibbs sampling conditional distribution for a topic indicator $z_{ij}$:

Figure 2: **STRADS LDA:** $\boldsymbol{s}$-error $\Delta_t$ at each iteration, on the Wikipedia unigram dataset with $K = 5000$ and 64 machines.

$$\mathrm{P}(z_{ij} \mid \boldsymbol{W}, \boldsymbol{Z}_{-ij}) \propto \mathrm{P}(w_{ij} \mid z_{ij}, \boldsymbol{W}_{-ij}, \boldsymbol{Z}_{-ij})\mathrm{P}(z_{ij} \mid \boldsymbol{Z}_{-ij})$$
$$= \frac{\gamma + B_{w_{ij},z_{ij}}}{V\gamma + \sum_{v=1}^{V} B_{v,z_{ij}}} \times \frac{\alpha + D_{i,z_{ij}}}{K\alpha + \sum_{k=1}^{K} D_{i,k}}.$$

In the first term, the denominator quantity $\sum_{v=1}^{V} B_{v,z_{ij}}$ is exactly the sum over the $z_{ij}$-th column of $\boldsymbol{B}$, i.e. $\boldsymbol{s}_{z_{ij}}$. Thus, errors in $\boldsymbol{s}$ induce errors in the probability distribution $U_{w_{ij}} \sim \mathrm{P}(w_{ij} \mid z_{ij}, \boldsymbol{W}_{-ij}, \boldsymbol{Z}_{-ij})$, which is just the discrete probability that topic $z_{ij}$ will generate word $w_{ij}$. As a proxy for the error in $U$, we can measure the difference between the true $\boldsymbol{s}$ and its local copy $\tilde{\boldsymbol{s}}^p$ on worker $p$. If $\boldsymbol{s} = \tilde{\boldsymbol{s}}^p$, then $U$ has zero error.

We can show that the error in $\boldsymbol{s}$ is empirically negligible (and hence the error in $U$ is also small). Consider a single STRADS LDA iteration $t$, and define its $\boldsymbol{s}$-error to be

$$\Delta_t = \tfrac{1}{PM} \sum_{p=1}^{P} \|\tilde{\boldsymbol{s}}^p - \boldsymbol{s}\|_1, \tag{1}$$

where $M$ is the total number of tokens $w_{ij}$. The $\boldsymbol{s}$-error $\Delta_t$ must lie in $[0, 2]$, where 0 means no error. Figure 2 plots the $\boldsymbol{s}$-error for the "Wikipedia unigram" dataset (refer to our experiments section for details), for $K = 5000$ topics and 64 machines (128 processor cores total). The $\boldsymbol{s}$-error is $\leq 0.002$ throughout, confirming that STRADS LDA exhibits very small parallelization error.

**Memory usage:** STRADS LDA explicitly partitions the model parameters into subsets, allowing us to solve big model LDA problems with limited memory. Figure 3 shows the memory usage of STRADS and YahooLDA [1] for topic modeling on unigram Wikipedia data with $10K$ topics. As shown in the figure, as the number of machines increases, STRADS used less memory per machine, but YahooLDA's memory usage decreased only slightly. In data-parallel YahooLDA, each worker stores a portion of word-topic table referenced by the words in the worker's data partition. However,

Figure 3: **Topic modeling: Memory usage per machine**, for model-parallellism (STRADS) vs data-parallellism (YahooLDA).

```
// STRADS Matrix Factorization
```

```
schedule() {
  // Round-robin scheduling
  if counter <= U      // Do W
    return W[q_counter]
  else                 // Do H
    return H[r_(counter-U)]
}
```

```
push(worker = p, pars = X[s]) {
  z = []              // Empty list
  if counter <= U   // X is from W
    for row in s, k=1..K
      z.append( (f_1(row,k,p),f_2(row,k,p)) )
  else                // X is from H
    for col in s, k=1..K
      z.append( (g_1(k,col,p),g_2(k,col,p)) )
  return z
}
```

```
pull(workers=[p], pars=X[s], updates=[z]) {
  if counter <= U   // X is from W
    for row in s, k=1..K
      W[row,k] = f_3(row,k,[z])
  else                // X is from H
    for col in s, k=1..K
      H[k,col] = g_3(k,col,[z])
  counter = (counter mod 2*U) + 1
}
```

Figure 4: **STRADS MF pseudocode.** Definitions for $f_1, g_1, \ldots$ and $q_p, r_p$ are in the text. `counter` is a global model variable.

in big model settings, STRADS's dynamic model partitioning strategy used memory more efficiently than YahooLDA's static model partitioning strategy.

## 2 Full STRADS Program for Matrix Factorization (MF)

STRADS model-parallel programming can be applied to matrix factorization (collaborative filtering); MF is used to predict users' unknown preferences, given their known preferences and the preferences of others. While most MF implementations tend to focus on small decompositions with rank $K \approx 100$ [14, 4, 13], we are interested in enabling larger decompositions with rank $> 1000$, where the much larger factors (billions of parameters) pose a challenge for purely data-parallel algorithms (such as naive SGD) that need to share all parameters across all workers; again, STRADS addresses this by explicitly dividing parameters across workers.

Formally, MF takes an incomplete matrix $\mathbf{A} \in \mathbb{R}^{N \times M}$ as input, where $N$ is the number of users, and $M$ is the number of items/preferences. The idea is to discover rank-$K$ matrices $\mathbf{W} \in \mathbb{R}^{N \times K}$

and $\mathbf{H} \in \mathbb{R}^{K \times M}$ such that $\mathbf{WH} \approx \mathbf{A}$. Thus, the product $\mathbf{WH}$ can be used to predict the missing entries (user preferences). Formally, let $\Omega$ be the set of indices of observed entries in $\mathbf{A}$, let $\Omega^i$ be the set of observed column indices in the $i$-th row of $\mathbf{A}$, and let $\Omega_j$ be the set of observed row indices in the $j$-th column of $\mathbf{A}$. Then, the MF task is defined as an optimization problem:

$$\min_{\mathbf{W}, \mathbf{H}} \sum_{(i,j) \in \Omega} (a_j^i - \mathbf{w}^i \mathbf{h}_j)^2 + \lambda(\|\mathbf{W}\|_F^2 + \|\mathbf{H}\|_F^2). \tag{2}$$

This can be solved using parallel CD [13], with the following update rule for $\mathbf{H}$:

$$(h_j^k)^{(t)} \leftarrow \frac{\sum_{i \in \Omega_j} \left\{ r_j^i + (w_k^i)^{(t-1)} (h_j^k)^{(t-1)} \right\} (w_k^i)^{(t-1)}}{\lambda + \sum_{i \in \Omega_j} \left\{ (w_k^i)^{(t-1)} \right\}^2}, \tag{3}$$

where $r_j^i = a_j^i - (\mathbf{w}^i)^{(t-1)} (\mathbf{h}_j)^{(t-1)}$ for all $(i,j) \in \Omega$, and a similar rule holds for $\mathbf{W}$.

**STRADS implementation:** Let us start with $\mathbf{R} = \mathbf{A}$ assuming that $\mathbf{W}$ and $\mathbf{H}$ are initialized with zeros. MF **schedule** strategy is to partition the rows of $\mathbf{R}$ into $U$ disjoint index sets $\{q_p\}_{p=1}^U$, and the columns of $\mathbf{R}$ into $U$ disjoint index sets $\{r_p\}_{p=1}^U$. Further, $\mathbf{W}$ and $\mathbf{H}$ are partitioned by the row index set $\{q_p\}_{p=1}^U$ and the column index set $\{r_p\}_{p=1}^U$, respectively. Figure 5 shows the partition scheme of $\mathbf{R}$, $\mathbf{W}$ and $\mathbf{H}$ given three workers ($U = 3$). We then dispatch the model parameters $\mathbf{W}, \mathbf{H}$ in a round-robin fashion, that is, $\mathbf{W}^{q_1}, \ldots, \mathbf{W}^{q_U}, \ldots, \mathbf{H}_{r_1}, \ldots, \mathbf{H}_{r_U}$ are updated in turn. Specifically, the **push** update for $\mathbf{H}$ in the $p$-th worker (the case for $\mathbf{W}$ is similar) computes

Figure 5: The partitioning scheme of STRADS MF given three workers. Shaded blocks are stored in each worker.

$$(o_j^k)_p^{(t)} \leftarrow g_1(k, j, p) := \sum_{i \in (\Omega_j)_p} \left\{ r_j^i + (w_k^i)^{(t-1)} (h_j^k)^{(t-1)} \right\} (w_k^i)^{(t-1)}, \tag{4}$$

$$(b_j^k)_p^{(t)} \leftarrow g_2(k, j, p) := \sum_{i \in (\Omega_j)_p} \left\{ (w_k^i)^{(t-1)} \right\}^2, \tag{5}$$

where $(\Omega_j)_p$ are the (observed) elements of column $\mathbf{r}_j$ indexed by $q_p$. Finally, **pull** aggregates the updates:

$$(h_j^k)^{(t)} \leftarrow g_3(k, j, [(o_j^k)_p^{(t)}, (b_j^k)_p^{(t)}]) := \frac{\sum_{p=1}^U (o_j^k)_p^{(t)}}{\lambda + \sum_{p=1}^U (b_j^k)_p^{(t)}},$$

with a similar definition for updating $\mathbf{W}$ using $(w_k^i)^{(t)} \leftarrow f_3()$ and $f_1(i, k, p)$, $f_2(i, k, p)$. This **push-pull** scheme is free from parallelization error. When $\mathbf{W}$ are updated by **push**, they are mutually independent because $\mathbf{H}$ is held fixed, and vice-versa.

## 3 Analysis of STRADS Lasso Scheduling for Parallel Coordinate Descent

Lasso [11] takes the following form of an optimization problem:

$$\min_{\boldsymbol{\beta}} \frac{1}{2} \|\mathbf{y} - \mathbf{X}\boldsymbol{\beta}\|_2^2 + \lambda \sum_j |\beta_j|, \tag{6}$$

where $\mathbf{X} \in \mathbb{R}^{N \times J}$ is the input data for $J$ inputs and $N$ samples, $\mathbf{y} \in \mathbb{R}^{N \times 1}$ is the output vector, $\boldsymbol{\beta} \in \mathbb{R}^{J \times 1}$ is the vector of regression coefficients, and $\lambda$ is a regularization parameter that determines the sparsity of $\boldsymbol{\beta}$. We solve (6) using a parallel coordinate descent (CD) algorithm [3] with a scheduler that dynamically finds a set of coefficients to be updated at runtime. Here we present the analysis of our scheduling schemes implemented under STRADS for parallel CD Lasso.

Let us start with the description of how our Lasso scheduling works under STRADS primitives:

1. **Sampler:** Select $L'(> L)$ indices of coefficients in the $t$-th iteration, following the distribution of $p(j) \propto \left( \delta\beta_j^{(t-1)} \right)^2 + \eta$, where $\delta\beta_j^{(t-1)} = \beta_j^{(t-2)} - \beta_j^{(t-1)}$ and $\eta$ is a small positive constant. We denote the set of selected $L'$ indices of coefficients by $\mathcal{C}$.

2. **Correlation checker:** From $\mathcal{C}$, randomly select $L$ indices of coefficients, denoted by $\mathcal{B}$, that satisfy $|\mathbf{x}_j^T\mathbf{x}_k| < \rho$ for all $j \neq k$ $j, k \in \mathcal{C}$. Here $\mathbf{x}_j^T\mathbf{x}_k$ represents the correlation between $\mathbf{x}_j$ and $\mathbf{x}_k$ because $\mathbf{X}$ is standardized (i.e., $\sum_i x_j^i = 0$, $\sum_i (x_j^i)^2 = N, \forall j$).

For analysis, we rewrite problem (6) by duplicating original features with opposite sign:

$$\min_{\boldsymbol{\beta}} \frac{1}{2} \|\mathbf{y} - \mathbf{X}\boldsymbol{\beta}\|_2^2 + \lambda \sum_{j=1}^{2J} \beta_j. \tag{7}$$

Here, with an abuse of notation, $\mathbf{X}$ contains $2J$ features and $\beta_j \geq 0$, for all $j = 1, \ldots, 2J$. We define $F(\boldsymbol{\beta}^{(t)}) = \frac{1}{2} \|\mathbf{y} - \mathbf{X}\boldsymbol{\beta}^{(t)}\|_2^2 + \sum_{j=1}^{2J} \beta_j^{(t)}$, and the following analysis shows that $p(j) \propto \left(\delta\beta_j^{(t)}\right)^2$ aims to increase Lasso convergence rate at every iteration.

**Proposition 1.** *Suppose $\mathcal{B}$ is the set of indices of coefficients updated in parallel at the $t$-th iteration, and $\rho$ is sufficiently small constant such that $\rho\delta\beta_j^{(t)}\delta\beta_k^{(t)} \approx 0$, for all $j \neq k \in \mathcal{B}$. Then, the sampling distribution $p(j) \propto \left(\delta\beta_j^{(t)}\right)^2$ approximately maximizes a lower bound on $E_{\mathcal{B}}\left[F(\boldsymbol{\beta}^{(t)}) - F(\boldsymbol{\beta}^{(t)} + \Delta\boldsymbol{\beta}^{(t)})\right]$.*

*Proof.* Let us denote $\mathcal{L}$ by a lower bound on $E_{\mathcal{B}}\left[F(\boldsymbol{\beta}^{(t)}) - F(\boldsymbol{\beta}^{(t)} + \Delta\boldsymbol{\beta}^{(t)})\right]$, that is,

$$\mathcal{L} \leq E_{\mathcal{B}}\left[F(\boldsymbol{\beta}^{(t)}) - F(\boldsymbol{\beta}^{(t)} + \Delta\boldsymbol{\beta}^{(t)})\right]. \tag{8}$$

Below, we show that $p(j) \propto \left(\delta\beta_j^{(t)}\right)^2$ approximately maximizes $\mathcal{L}$. We start with the assumption used in [3]:

$$F(\boldsymbol{\beta}^{(t)}) - F(\boldsymbol{\beta}^{(t)} + \Delta\boldsymbol{\beta}^{(t)}) \geq -(\Delta\boldsymbol{\beta}^{(t)})^T\nabla F(\boldsymbol{\beta}^{(t)}) - \frac{1}{2}(\Delta\boldsymbol{\beta}^{(t)})^T\mathbf{X}^T\mathbf{X}(\Delta\boldsymbol{\beta}^{(t)}), \tag{9}$$

where $(\Delta\boldsymbol{\beta}^{(t)})^T\nabla F(\boldsymbol{\beta}^{(t)}) \leq 0$. For simple notation, let us omit the superscript representing the $t$-th iteration.

Suppose that the index of coefficient $j$ is drawn from a distribution $p(j)$, and a pair of indices $(j, k)$ is drawn from $p(j, k)$. Taking expectation of (9) with respect to $\mathcal{B}$, we have

$E_{\mathcal{B}}[F(\boldsymbol{\beta}) - F(\boldsymbol{\beta} + \Delta\boldsymbol{\beta})]$

$$\geq -E_{\mathcal{B}}[\sum_{j\in\mathcal{B}} \delta\beta_j \nabla(F(\boldsymbol{\beta}))_j] - \frac{1}{2}E_{\mathcal{B}}\left[\sum_{\{(j,k):j,k\in\mathcal{B}\}} \delta\beta_j(\mathbf{x}_j^T\mathbf{x}_k)\delta\beta_k\right] \tag{10}$$

$$= -L\sum_{j\in\mathcal{B}} p(j)\left[\delta\beta_j\nabla(F(\boldsymbol{\beta}))_j + \frac{1}{2}(\delta\beta_j)^2\right] - \frac{1}{2}L(L-1)\left[\sum_{\{(j,k):j,k\in\mathcal{B},j\neq k\}} p(j,k)\delta\beta_j(\mathbf{x}_j^T\mathbf{x}_k)\delta\beta_k\right] \tag{11}$$

$$= -L\sum_{j\in\mathcal{B}} p(j)\left[\delta\beta_j\nabla(F(\boldsymbol{\beta}))_j + \frac{1}{2}(\delta\beta_j)^2\right] - \frac{1}{2}L(L-1)\left[\sum_{\{(j,k):j,k\in\mathcal{B},j\neq k,|\mathbf{x}_j^T\mathbf{x}_k|<\rho\}} p(j,k)(\mathbf{x}_j^T\mathbf{x}_k)\delta\beta_j\delta\beta_k\right] \tag{12}$$

$$\approx L\sum_{j\in\mathcal{B}} p(j)\left[-\delta\beta_j\nabla(F(\boldsymbol{\beta}))_j - \frac{1}{2}(\delta\beta_j)^2\right]. \tag{13}$$

In (11), we used $p(j, k) = 0$ if $|\mathbf{x}_j^T\mathbf{x}_k| \geq \rho$ because $\beta_j$ and $\beta_k$ cannot be updated in parallel if $|\mathbf{x}_j^T\mathbf{x}_k| \geq \rho$. Recall that we find $\mathcal{B}$ such that $|\mathbf{x}_j^T\mathbf{x}_k| < \rho$ for all $j \neq k, j, k \in \mathcal{B}$ using the correlation checker. In (12), we used our assumption that $\rho\delta\beta_j\delta\beta_k \approx 0$ for all $j \neq k \in \mathcal{B}$. From (13), we can see that the lower bound of $E_{\mathcal{B}}[F(\boldsymbol{\beta}) - F(\boldsymbol{\beta} + \Delta\boldsymbol{\beta})]$ is maximized when

$$p(j) \propto \left| -\delta\beta_j\nabla(F(\boldsymbol{\beta}))_j - \frac{1}{2}(\delta\beta_j)^2 \right|. \tag{14}$$

Now let us consider the following CD update rule: $\delta\beta_j = \max\{-\beta_j, -\nabla(F(\boldsymbol{\beta}))_j\}$ [3]. Based on this update rule, we have the following two cases: If $\delta\beta_j = -\nabla(F(\boldsymbol{\beta}))_j$,

$$p(j) \propto \left| -\delta\beta_j\nabla(F(\boldsymbol{\beta}))_j - \frac{1}{2}(\delta\beta_j)^2 \right| = \frac{1}{2}(\delta\beta_j)^2.$$

If $\delta\beta_j = -\beta_j$,

$$p(j) \propto \left| -\delta\beta_j \nabla(F(\boldsymbol{\beta}))_j - \frac{1}{2}(\delta\beta_j)^2 \right| \geq \frac{1}{2}(\delta\beta_j)^2.$$

Here we used the fact that $\delta\beta_j \geq -\nabla(F(\boldsymbol{\beta}))_j$ due to the CD update rule, and $\delta\beta_j \leq 0$ because $\beta_j \geq 0$, as defined by (7). It shows that $p(j) \propto \frac{1}{2}(\delta\beta_j)^2$ approximately maximizes the lower bound $\mathcal{L}$ of $E_{\mathcal{B}}[F(\boldsymbol{\beta}) - F(\boldsymbol{\beta} + \Delta\boldsymbol{\beta})]$.

$\square$

## 4 Discussion and Related Work

As a programmable framework for dynamic Big Model-parallelism, STRADS provides the following benefits: (1) scalability and efficient memory utilization, allowing larger models to be run with additional machines (because the model is partitioned, rather than duplicated across machines); (2) the ability to invoke dynamic **schedules** that reduce model parameter dependencies across workers, leading to lower parallelization error and thus faster, correct convergence.

While the notion of model-parallelism is not new, our contribution is to study it within the context of a programmable system (STRADS) that enables managed scheduling of parameter updates (based on model dependencies). Previous works explore aspects of model-parallelism in a more specific context: Scherrer et al. [10] proposed a static model partitioning scheme specifically for parallel coordinate descent, while GraphLab [8, 9] statically pre-partitions data and parameters through a graph abstraction.

An important direction for future research is to reduce the communication costs of using STRADS. Currently, STRADS adopts a star topology from scheduler machines to workers, which causes the scheduler to eventually become a bottleneck as we increase the number of machines. To mitigate this issue, we wish to explore different **sync** schemes such as an asynchronous parallelism [1] and stale synchronous parallelism [7]. We also want to explore the use of STRADS for other popular ML applications, such as support vector machines and logistic regression.