[Reviews · NeurIPS 2014]

Submitted by Assigned_Reviewer_33

This paper introduces a set of primitives for parallelizing models too large to fit in memory. The primitives are (1) schedule; which determines which subset of parameters will be assigned to each machine, (2) push; which computes a partial model update in a given worker, (3) pull; which aggregates the partial results from workers. These functions should be implemented by the user. Synchronization doesn't need to be implemented by the user: the system uses a built-in primitive based on bulk synchronous parallel syncing (ensuring all workers access up-to-date parameters). The system is applied to solve instances of LDA, matrix factorization and Lasso. Empirical comparisons with YahooLDA and GraphLab show faster convergence and improved runtime. Scalability is shown to be close to linear.

The paper is clearly written, the primitives are intuitive, the empirical results are satisfactory. Overall a good paper.

I have a few questions:

Is there a good reason why no comparison has been made against the parameter server from http://parameterserver.org ? This seems to directly address the issue of model parallelism that you focus on in this paper

Your sync seems to assume workers won't fail. How would you scale the framework to thousands of machines? Wouldn't the proposed synchronization strategy become too naive in that case since worker failure becomes an issue? I.e. how to make the architecture more fault tolerant?
Summary: The paper introduces a framework of primitives for model parallelization and applies it to a number of learning problems. Experimental results comparing it against a few recent approaches for large-scale learning show improved runtime and faster convergence.

Submitted by Assigned_Reviewer_36

The authors proposed a framework of primitives for model parallelism, in order to address the problem that a single machine may be inadequate to fit a big model in memory.

This seems to be an interesting paper, well written and organized, albeit some small typos are found (e.g., q_idx -> idx in Figure 4, 1/L -> 1/P in line 140).

Toward the significance of this paper, I have the following concerns:

1. I agree that for models such as deep neural networks (e.g. CNN), more parameters often leads to better performance, as demonstrated empirically by [5] (using the paper's bib reference number), which justifies the need of more efficient model parallelism framework for bigger models. However, for models like topic models (LDA) and matrix factorization (MF), I am skeptical toward such need. For example, would LDA with more than tens of thousands of topics, or MF with rank larger than a few thousands, lead to better model performance, compared against their counterparts with moderate number of parameters? The work could be more interesting if the authors address this issue.

Besides, can the proposed framework be applied to models that already proved to enjoy more parameters, such as deep neural networks? If yes, how would it perform when compared to the one proposed in [5]? In short, I think more compelling use cases for the proposed framework would greatly enhance the potential impact of this work on the NIPS community.

2. The experiments.
1) Some important specifications for the experiments are missing. On the evaluation metrics for LDA and MF, how was the convergence being measured? Take LDA, was the log-likelihood calculated on training documents or testing (unseen) documents? One would imagine LDA with huge number of topics can easily over-fit, if no special regularization or sparsity-inducing priors being adopted, and such convergence could be meaningless. And how was the memcached (which is one of the keys for the scalability for YahooLDA) configured? And for the proposed framework, how were the summary statistics D and B stored and distributed to the work nodes, as their size grow proportionally to the number of parameters? These are just some key factors that may need to be clarified, as they may contribute to the difference observed in Figure 8 and 9.

2) The comparison between STRADS and Graphlab in the MF experiment is somehow unfair. Because STRADS used the cyclic coordinate descent (CCD) as its optimization algorithm, whereas Graphlab used alternating least square (ALS). As it has been demonstrated in [21], CCD converges faster than ALS and is also more scalable. As a result, it would be more interesting to compare STRADS against Graphlab with the *same* optimization algorithm. Furthermore, the method proposed in [9] is related, and would also be an interesting baseline for comparison. With such additional experimental results, one may gain further insight into the STRADS framework.

Summary: This paper is well written. However, the use cases of the proposed framework demonstrated in this paper are not entirely convincing, and the experiments lack some necessary and important details.

Submitted by Assigned_Reviewer_42

The authors introduce a new framework (programmable system) for parallel computation with an emphasis on big model problems: learning tasks where the parameters themselves (not just the data) are too many to fit in a single machine's memory. It relies on simple primitives similar to the ones found in existing systems with one important distinction.
The "schedule" primitive, which is a programmable function that dynamically decides the scheduling: the assignment of specific subsets of the parameters to workers. The workers then work on a subset of the data to update a subset of the parameters. The other primitives take care of aggregation of all the updates. This kind of dynamic scheduling makes it possible to:
1. Focus on updating the parameters that seem more promising (changing at a greater rate).
2. Avoid updating parameters with dependencies in parallel. This also improves the convergence behavior.

Pros:
The idea behind this work is intuitive, novel (to the best of my knowledge) and well motivated: static scheduling used in current systems is far from optimal. Dynamic and problem-dependent scheduling sounds - and seems to be in experiments - very promising.

The writing is very good and the presentation of the algorithms in pseudocode makes the paper very readable.

The authors implemented algorithms for and experimented extensively on three major problems. The results of their evaluation show promise for their method.

Cons:

1. As the authors mention in the conclusion, the parameters are stored and need to be communicated to worker nodes at every round. Requiring the full set of parameters to be communicated the worker nodes at every round sounds very costly and could potentially be avoided.

2. Related to the above point: in the selection process for Lasso (top of page 7), the scheduler(s) need (seemingly) access to all data points to evaluate inner products. That seems excessive. What is the cost (communication/computation) of that? Would the need to study any data point in scheduling unnecessarily inflate the number of schedulers?

3. The choice of rank greater than 1000 for the matrix factorization examples seems excessive and potentially implies that is the region where the authors were able to show improvement over the state of the art. This is not necessarily a problem; just this region doesn’t seem so well motivated.
Summary: Well motivated problem and solution, extensive experiments on three important problem and good potential for impact. I recommend acceptance.

Submitted by Assigned_Reviewer_43

The author proposed a new large scale machine learning framework. It claims to improve the current approaches, for its ability to do partial model updates and to dynamically schedule model updates based on feature importance or convergence speed.

However, the proposed scheduling interface is not suitable for large scale machine learning. Because it requires accessing all model variables and returning the whole list of variables be updated in a single function call. This is problematic because large scale model size can easily reach the order of 10^12, exceeding the single machine capacity. And in this case, how to coordinate dynamic scheduling among nodes becomes unclear, which is supposed to be the key contribution of the paper.

Also for push and pull interface, the update from a single worker can be huge, which should not be returned or processed in a single function. The current interface will limit the number of variables in a single update. A key-value based scheme is the industry standard.

Finally, when comparing to existing systems, graphlab is capable of this kind scheduling through node signaling, although extra effort may be required.
Summary: The proposed system lacks any real test of industrial level large-scale learning problems, and this resulted in inappropriate choices of framework design.
Author Feedback
Author rebuttal: We thank the reviewers for their enthusiastic response and valuable suggestions. This paper is one part of our efforts to develop a massively scalable ML system on 1000s of machines (as one reviewer mentioned). Such a system requires not only intricate system architecture and protocol design, but also a deep understanding of the characteristics, robustness, and decomposability of machine learning programs, which, in our view, seem to go undiscussed in many recent Big-ML framework papers (which are mostly focused on exploiting data-parallelism). Our paper is focused on closely examining the properties of typical but non-trivial ML programs (which we believe are representative of broad ML families), and thereupon define operation primitives that can best leverage ML properties to facilitate sound, implementable, and fine-grained model-parallelism. Admittedly, our work does not yet address certain issues in distributed ML, such as fault tolerance at 1000s of machines, and we see our work as exploring a different yet complementary direction to parameter servers. We hope this will lead to new insights that can speed up both medium- and large-scale ML problems, while being memory-efficient enough to allow larger models to be solved on current hardware.

All:

-Why do we need big LDA, MF and Lasso models?

LDA with many topics (>10^4) significantly improves performance [Wang et al. 2014, http://arxiv.org/pdf/1405.4402v1.pdf]. Wang et al. reported that for industrial-scale big data, big topic models are desirable because many topics can cover long-tail semantic word sets in such big data. For example, in online advertising, the ad click-through rate prediction accuracy was substantially increased going from 10^4 to 10^5 topics. Such applications typically go beyond extracting only topics for human interpretation or visualization.

In MF, for Big Data matrices with high intrinsic complexity (e.g., many movie types and user groups), low rank would lead to severe loss of information, as can be seen in a plot of the eigenvalues or the reconstruction error using low rank basis. Zhou et al. (2008) reported that high-rank L2-regularized MF (like in STRADS) yields improved performance up to rank 1000 on the Netflix dataset.

For Lasso, there exist very high-dimensional applications in genetics or finance. For example, 10-K Corpus [Kogan et al, 2009] contains >4*10^6 features. In genetics, for modeling interactions between genetic variants, the number of feature pairs (e.g. single nucleotide polymorphisms) to include in the Lasso model is often greater than 10^10.

Moreover, we view STRADS as not just useful for large ML models, but also as a platform to diagnose and improve ML at medium scales. For example, our lower-rank (K<100) MF results, also exhibit accelerated convergence over GraphLab. We believe that the speedup opportunities exposed by variable scheduling apply to a wide range of problem scales.

R1:

-Why is there no comparison to parameter server?

We did not compare STRADS with parameter servers (PSes) because STRADS and PSes have rather different approaches to accelerating ML; a truly fair comparison would demand work that is beyond the scope of this paper. STRADS adapts an ML algorithm via dynamic variable scheduling (similar to GraphLab’s rationale), making it converge in fewer iterations. On the other hand, PSes provide a synchronization service, instead of a scheduling service.

-Machine failure issue

Our focus is on the techniques to improve time to convergence of a specific model on a specific input. We usually restart from original data after a failure. With scaled up data sets, we would implement classic checkpoint-restart based on the state of each machine at the end of a schedule set. Moreover, checkpoint bandwidth will be low because the number of variables modified per computation is small compared to the total size of the problem. The application of STRADS techniques to online learning is a different and interesting problem which we are interested in studying as future work.

R2:

-Application of STRADS to neural networks

STRADS for neural networks is out of this paper’s scope but there can be scheduling opportunities for speedup. Dynamic partition of a large DNN into meaningful blocks (possibly other than just layerwise partition) and distribution of such blocks to workers for parallel learning offers another source for speeding up.

-Evaluation metrics for LDA and MF

We calculated the LDA log-likelihood and MF objective function on the training data. Our intent is to show that STRADS accelerates the convergence of ML algorithms during the training phase.

-Memcached configuration for YahooLDA

We used the newer WSDM’12 YahooLDA (rather than VLDB’10, which the reviewer is referring to), that no longer uses memcached, and insensitive to configuration.

-Storage/distribution of summary statistics D and B

We store both D (doc-topic) and B (topic-vocab) in a distributed fashion. Only B needs to be synchronized across all workers; for B we used a lightweight distributed key-value store. For D, we only kept a local partition in each worker.

-Different algorithms used in STRADS MF and GraphLab MF

CCD MF is not available in the latest GraphLab 2.2, so we compared against ALS MF instead. We were unable to compare STRADS MF with the method in [9] since the code for [9] is publicly unavailable.

R3:

-Cost of parameter sync

Parameter synchronization efficiency is not the focus of this paper, but as future work, we would augment STRADS with a parameter server for efficient parameter synchronization (e.g. Ho et al., 2013).

-Cost of evaluating scheduler inner products in Lasso

Computing inner products is not a bottleneck. Schedulers compute them for only subsampled features; further, we overlap communication and computation for efficiency. We always included scheduling machines in our machine counts, for fairness.